# Analysis of Gene Single Nucleotide Polymorphisms in COVID-19 Disease Highlighting the Susceptibility and the Severity towards the Infection

**DOI:** 10.3390/diagnostics12112824

**Published:** 2022-11-16

**Authors:** Mario Giosuè Balzanelli, Pietro Distratis, Rita Lazzaro, Van Hung Pham, Toai Cong Tran, Gianna Dipalma, Angelica Bianco, Emilio Maria Serlenga, Sergey Khachatur Aityan, Valentina Pierangeli, Kieu Cao Diem Nguyen, Francesco Inchingolo, Diego Tomassone, Ciro Gargiulo Isacco

**Affiliations:** 1SET-118, Department of Pre-Hospital and Emergency-San Giuseppe Moscati Hospital, 74100 Taranto, Italy; 2Department of Microbiology and Virology, Phan Chau Trinh University of Medicine, Danang City 50000, Vietnam; 3Department of Histology, Embryology and Genetics, Pham Ngoc Thach University of Medicine, Ho Chi Minh City 70000, Vietnam; 4Department of Interdisciplinary Medicine, University of Bari “Aldo Moro”, 70126 Bari, Italy; 5Experimental Zooprophylactic Institute of Puglia and Basilicata, 71121 Foggia, Italy; 6Hematology Department, Blood Transfusion Unit, SS Annunnziata Hospital, 74100 Taranto, Italy; 7Multidisciplinary Research Center, Lincoln University, Oakland, CA 94612, USA; 8NEFLT NHS Foundation Trust, Rainham RM13 8EU, UK; 9Foundation of Physics Research Center, 87053 Celico, Italy

**Keywords:** arterial blood gas (ABG), carbon dioxide (SpCO_2_), COVID-19, oxygen (SpO_2_), SARS-CoV-2

## Abstract

Many factors may influence the risk of being infected by SARS-CoV-2, the coronavirus responsible for coronavirus disease 2019 (COVID-19). Exposure to the virus cannot explain the variety of an individual’s responses to the virus and the high differences of effect that the virus may cause to some. While a person’s preexisting condition and their immune defenses have been confirmed to play a major role in the disease progression, there is still much to learn about hosts’ genetic makeup towards COVID-19 susceptibility and risk. The host genetic makeup may have direct influence on the grade of predisposition and outcomes of COVID-19. In this study, we aimed to investigate the presence of relevant genetic single nucleotide polymorphisms (SNPs), the peripheral blood level of IL6, vitamin D and arterial blood gas (ABG) markers (pH, oxygen-SpO_2_ and carbon dioxide-SpCO_2_) on two groups, COVID-19 (n = 41, study), and the healthy (n = 43, control). We analyzed cytokine and interleukin genes in charge of both pro-inflammatory and immune-modulating responses and those genes that are considered involved in the COVID-19 progression and complications. Thus, we selected major genes, such as IL1β, IL1RN (IL-1 β and α receptor) IL6, IL6R (IL-6 receptor), IL10, IFNγ (interferon gamma), TNFα (tumor necrosis factor alpha), ACE2 (angiotensin converting enzyme), SERPINA3 (Alpha-1-Antiproteinase, Antitrypsin member of Serpin 3 family), VDR (vitamin D receptor Tak1, Bsm1 and Fok1), and CRP (c-reactive protein). Though more research is needed, these findings may give a better representation of virus pleiotropic activity and its relation to the immune system.

## 1. Introduction

Individuals are differentially affected by COVID-19. While preexisting background disorders have been studied extensively, little is known about the genetic variation underlying this occurrence. The rapidity of transmission and the sudden clinical decay during severe acute respiratory syndrome coronavirus 2 (SARS-CoV-2) infection immediately appeared as peculiar traits of the disease. During the first and second wave of the Delta variant, organ failure and sepsis in COVID-19 were common events, and the death toll was extremely high. The excessive uncontrolled inflammatory responses mainly led by IL-6, IFNγ, TNFα, CRP, D-dimer, VES, and the lack of vitamin D (pre-hormone D) were common clinical traits of this infection, though those parameters were common to other types of lung diseases and infections. To make the scenario even more complicated, there was little or none homogeneity among patients and among the unpredictability of the events [1,2,3]. After an initial period of uncertainty, researchers started to assemble together information and data. The different investigative approaches drove crucial conclusions essential for predicting, diagnosing, and treating COVID-19 in all of its manifestations. One of the best achievement in this direction was to highlight the important role of single nucleotide polymorphisms (SNPs) of those genes involved in the immune regulatory mechanism. Extensive disproportion to virus susceptibility was ascribed by scientists to individuals’ genetic background and to environmental factors. In fact, the disease grade of severity was soon observed in connection to diverse gene carrying specific SNPs [3,4,5,6,7,8,9].

This is a retrospective, double-center observational study-control study, conducted at the 118 Pre-hospital and Emergency Department (Semi-intensive Care Unit) of SG Moscati Hospital of Taranto, Italy. The diagnosis of COVID-19 was performed by pulmonary computed tomography (CT-scan) and oropharyngeal swabs (RT-PCR) obtained from patients and healthy individuals recruited between September 2020 and October 2020. Patients admitted to the 118 Semi-intensive Care Unit revealed light, mild and severe symptoms, although all indistinctly confirmed a “ground glass” opacity by thoracic CT scan. A great part of those patients with mild symptomatology showed mild dyspnea, light fever, and dry cough. Few of them worsened almost suddenly and were admitted to sub-intensive care unit (ICU). Eventually, almost all of the patients were successfully treated, although many revealed persistent health problems even after infection (later called “long COVID-19”).

Thus, in the present study, we planned to investigate the presence of multiple SNPs that are generally considered to be involved in inflammatory and autoimmune disorders, providing an indication that changes in some of these gene regulatory sequences could be associated with different responses seen in the COVID-19 disease. The analyzed genes were the following: IL-1β (rs16944; rs1143634), IL-1RN (rs419598), IL-6 (rs1800796; rs1800795), IL-6R (rs2228145), IL-10 (rs1800896), IFNγ (rs2430561), TNFα (rs1800629), VDR vitamin D receptor (BsmI (G/A), FokI (C/T) e TaqI (T/C)), ACE-1 (rs4343), AACT-Serpin3 (rs1884082), and CRP (rs1205). Thus, accepting the importance of an individual’s genetic makeup involved in SARS-CoV-2 infection as our intent was eventually required to propose a different approach to applying tools of a personalized medicine. To this end, we analyzed and compared different SNPs of eleven genes known to be involved in the human immunity regulatory system and which play a key role in different system and tissue homeostases [9,10,11,12,13,14,15,16,17,18].

### 1.1. ACE Gene

The ACE 1 and 2 genes have been extensively investigated and were considered key contributors to SARS-CoV-2 infiltration into the system. The rs1799752 D/D genotype (deletion/deletion) was associated with an increased risk of cardiovascular disease due to the increase in plasma of ACE levels (double compared to subjects with genotype insertion/insertion-II). The rs1799752 I/D genotype (insertion/deletion) has been widely investigated in different inflammatory diseases with high frequency in cardio-vascular and respiratory diseases such as hypertension, STEMI, acute respiratory distress syndrome (ARDS), and pulmonary thromboembolism [19,20,21,22,23,24].

### 1.2. AACT-Serpina3 Gene

The Alpha1-antichimotrypsin (ACT: new identification—Serpina3) was considered. The Serpina3 gene plays an important modulating role in many inflammatory and immune modulatory processes. Conversely, Serpina3 polymorphisms seem to be tissue specific and have been considered to play a key role as an inflammatory promoter in neuro-degenerative diseases such as in Alzheimer’s disease and Parkinson’s disease. Serpina3 is a typical acute phase protein secreted into the circulation during acute and chronic inflammation. Its main target is likely neutrophil cathepsin G, a proinflammatory enzyme released at inflammation sites that contributes to the activation of inflammatory cytokines, pathogen degradation, and tissue remodeling. Cathepsin G promotes platelet aggregation. At the sites of vascular lesions and atherosclerosis, this enzyme is capable of converting angiotensin I into active angiotensin II. Furthermore, it increases the permeability of endothelial barriers by contributing to the migration and activation of perivascular lymphocytes. Therefore, by inhibiting cathepsin G, Serpina3 should limit inflammation, coagulation, remodeling of the ECM, and should inhibit apoptosis [25,26,27,28].

### 1.3. CRP Gene

Circulating CRP concentration is an explicit reflection of systemic inflammation and also actively participates in the physio-pathology of different diseases. C-reactive protein shows high expression during inflammatory conditions such as rheumatoid arthritis, some cardiovascular diseases, and infection. For instance, polymorphisms of CRP 3872 G-T G at rs1205 were associated with elevated CRP level associated with diabetes mellitus type 2 (DM2), cardiovascular diseases (CVD), and all the group of neuropathies related to metabolic disorders. Plasma levels of C-reactive protein are seen to be induced by the SARS-CoV-2 virus. The increase in the CRP concentration, as well as the onset concentration, would be indicative of clinical decay often followed by multi-organ injuries and massive uncontrolled inflammation that drives tissue fibrosis even months later after the infection. CRP-mediated neutrophil macrophage activation (M1) is suspected to be the main cause of pulmonary fibrosis and subsequent organ failure in COVID-19. Plasma levels of CRP, induced by severe acute respiratory syndrome coronavirus-2 (SARS-CoV-2) triggering COVID-19, can rise surprisingly high [29,30,31,32].

### 1.4. IL-10 Gene

The IL-10 gene polymorphisms (rs1800896) that limit or decrease the expression of IL10, are implicated in susceptibility to pulmonary infection and inflammation, such as tuberculosis (PTBC) and ARDS, specifically in the adult and elderly. Data showed that, among patients with ARDS, the G/G genotype with normal expression of IL10 revealed a decrease in severity with lower systemic failures and lower mortality rates. Conversely, data showed that both the G/A and the A/A genotypes are indicative of a lower IL-10 expression revealing higher susceptibility to COVID-19 affection. This suggests that these genotypes make IL-10 fail in controlling immune responses, either by allowing uncontrolled expression of proinflammatory cytokines such as IL-6 and TNFα, or by facilitating auto-aggression and external invasion [32,33,34].

### 1.5. IL-6 Gene

The polymorphisms in IL-6 genes are sadly notorious for procuring the “cytokine storm” as the main cause of deadly collaterals in the acute phase of the COVID-19 disease. The two major forms of IL-6 are the C-572G (rs 1800796) and G174C (rs1800795) polymorphisms. The IL-6 572 is involved in anti-E production, with the allele G as the major risk (the anti-E are antibodies directed against the AB0 system, targeting either A or B). Similarly, the *IL-6* 174 gene polymorphism is responsible for inflammatory and neuropathic traits in diseases such as DM2, atherosclerosis, and CVD. Though little is known regarding IL-6 polymorphisms and the pathogenesis of idiopathic pulmonary fibrosis in SARS-CoV-2 infections, data tend to confirm that the IL-6 174 is associated with elevated inflammatory patterns in the outcome of SARS-CoV-2 pneumonia. This event could be explained by the direct impact on CD4 and CD8 T cell fate exerted by IL-6 174 [35,36,37,38,39,40,41].

### 1.6. IL-1 Gene

The SNP rs16944 C/T in IL1β gene related to the promoter region has been associated with increased IL1β production with a consequent increased risk of developing inflammatory diseases and gastric carcinoma. Similarly, the SNP rs1143634 C/T has a role during aggressive periodontitis as an inhibitory factor for the expression of IL-10 [42,43,44,45].

### 1.7. IL-1 RN Gene

*IL1RN* belongs to the IL1 family. Its gene variants were seen involved in several degenerative condition either due to over-expression or down-expression of IL-1β. IL-1 RN is well known for its strong immune modulator capabilities. Any dysregulation may elicit the overproduction of proinflammatory cytokines and chemokines due to gene down-expression or, conversely, may lead to immune suppression as a consequence of gene overexpression, resulting in the enhancement of immune-mediated pathology [42].

### 1.8. IL-6 R Gene

The SNP rs2228145 A/C in IL-6 R gene has been associated with lower circulating inflammation biomarkers in coronary artery disease (CAD), such as C-reactive protein (CRP), IL-6, and fibrinogen. IL-6 R may control and modulate the pleiotropic activities of IL-6 through the membrane-bound and soluble forms, known as classic and trans-signaling, respectively. Of note, IL-6 R is primarily expressed by microglia in the central nervous system (CNS), which allowed scientists to hypothesize that alterations of the D358A variant in the ratio of transmembrane to soluble IL-6R may lead to genetic changes detectable in neurodegenerative diseases [12,13,14,15,16,17,36,37,38,43,44,45].

### 1.9. TNFα Gene

Tumor necrosis factor-alpha (TNFα) is an important cytokine and has been reported to be associated with the pathogenesis of many autoimmune and inflammatory diseases. The SNP rs1800629 was associated with high level of CVD, atherosclerosis, and high susceptibility to pulmonary disease risk. The presence of SNPs in a region of the TNF-α gene promoter tends to an uncontrolled increase in the expression of TNF-α at the systemic level in response to pathogenic insults, leading to immune dysregulations and organ damage. The TNF-α gene dysregulation is often associated with a high-level inflammatory process in many conditions other than COVID-19, such as CVD, atherosclerosis, and high susceptibility to lung disease risk [46,47,48,49].

### 1.10. IFNγ Gene

Eukaryotes cells, as well as mature somatic cells, are equipped with an IFNγ immune mechanism extremely important against external pathogen invasion. Functional studies have demonstrated that the SNP rs2430561 can increase or decrease the risk of a pathogenic infection and susceptibility for pulmonary infection such as tuberculosis, SARS, and oral infection (periodontitis). The IFNγ (1 and 2) is the main defensive mechanism against pathogen and microorganism invasion adopted by adult somatic cells. The IFNγ anti-viral function is based on extracellular and intracellular process by disrupting replication mechanisms via the inhibitory effect on virus gene expression and translation by impeding nucleocapsid assembly; the IFNγ breaks the disulfide bond impeding cell-to-cell interaction and suppresses the transcription activity of a virus. The presence of SNPs may either inhibit IFNγ anti-pathogen activity or increase its expression and, thus, leads to uncontrolled immune responses [50,51,52].

### 1.11. VDR Gene

Growing evidence documented the influence of vitamin D in the prognosis of COVID-19 infected patients. The vitamin D effectiveness and functionality strictly depends on the vitamin D receptor (VDR) gene within vitamin D absorption and the receptor’s signaling pathway. The overall damages caused by the hyper-expression of proinflammatory cytokines and interleukins, such as IL-6, may facilitate a factual decrease in the level of vitamin D due to the presence of SNPs at the level of the VDR gene. The reduced presence of vitamin D due to SNPs’ main expression sequences was often found together with a reduced functionality immune responses and decreased renal activity, low eGFR, and an increased mortality risk in patients with preexisting lung disorders and heart failure [52,53,54,55].

## 2. Materials and Methods

### 2.1. Study Design

This is a retrospective, double-center observational study-control study conducted at the 118 Pre-hospital and Emergency Department of SG Moscati Hospital of Taranto, Italy. The diagnosis of COVID-19 was performed by pulmonary computed tomography (CT-scan) and oropharyngeal swabs from patients and healthy individuals recruited between September 2020 and October 2020. The study was conducted in compliance with recognized international standards and the principles of the Declaration of Helsinki. This study has received the approval of The Independent Medical Ethics Committee of Brindisi, Protocol N. 44941-R.C.E. 81/20.

The genetic analysis was performed on saliva samples collected from 84 consent patients divided into two groups: a study group diagnosed with COVID-19 (n = 41; males n = 29; females n = 12; median age = 54.5) and a control group of healthy individuals which were negative to the RT-PCR COVID-19 test without clinical symptoms of lung infection (n = 43; males n = 16; females n = 27; median age = 36.5). The patients recruited in the control group were patients who received anti-inflammatory therapy consisting of vitamin D, vitamin K2, and aspirin. From our data (which have not been published), during that period, none of those patients were hospitalized for COVID-19 infection. There was significant difference in age and sex distributions in the groups (*p* < 0.05). Each individual within the COVID-19 and healthy groups was measured and assigned based on specific criteria evaluated at admission, such as fever, dyspnea, arterial blood gas analysis (ABG), oral-nasopharyngeal swab/RT-PCR, and thoracic CT-scan. The COVID-19 group comprised individuals that showed fever, dyspnea (ABG = pO2 < 60), a confirmed positive result by oral-nasopharyngeal swab/RT-PCR, and CT-scan showing ground–glass opacities. The healthy group included all individuals tested negative to nasopharyngeal swab/RT-PCR and normal ABG values. Data and findings were collected and compared [5]. It used a special kit (the swab from GenomaDiagnostic^®^) explicitly studied for the detection and the diagnosis for the presence of SNPs by high selective characterization of differences in mucosa. Sampling was performed by applying the procedures described in the kit (GenomaDiagnostic^®^, Rome, Italy), rolling the swab in the buccal area, and removing the saliva contents. The collected samples were sealed into the deep containers and sent to the Genoma laboratory for processing.

### 2.2. RT-PCR

The molecular test was conducted by carrying out the analysis of polymorphisms (Table 1), as indicate above. For the genotyping of the aforementioned polymorphisms, a DNA amplification by enzymatic reaction was carried out with polymerase chain reaction (PCR). DNA isolation was performed on saliva samples, and DNA was extracted by using a QIAamp DNA Mini Kit (Qiagen, Hilden, Germany), according to the manufacturer’s protocol. Quality and quantity of the isolated DNA were measured by nanodrop (ND-1000, Thermo Fischer Scientific, Wilmington, DE, USA). Analysis of SNP allele-specific SNP type assays was performed using a Fluidigm Flex Six™ Genotyping IFC (Fluidigm Corp., South San Francisco, CA, USA). Specific target amplification (STA) was performed to increase the number of molecular targets at the beginning. The determined thermal cycle program was run on a Bioer Gene Pro thermal cycler (95 °C for 15 min, followed by 14 cycles of 95 °C for 15 s and 60 °C for 4 min). SNP type assay mixes and sample mixes were prepared according to the manufacturer’s protocol. After loading a dynamic array with 4 μL of each 10× assay mix and 5 μL of each sample mix, it was placed on an IFC Controller HX (Fluidigm), and the loading process was completed. The dynamic array was then placed in the BioMark system (Fluidigm), which performs the thermal cycling and fluorescent image acquisition. The build-in data collection software of the BioMark system was used. Genotyping application, ROX passive reference, SNPtype-FAM, and SNPtype-HEX probe types were selected. The SNPtype E Flex Six v1 protocol was used for thermal cycling and image capture. The genotypes of the samples were subjected to automated sequence analysis using an automatic sequencer with fluorescent technology (ABI PRISM 3100 Genetic Analyzer). The mutation analysis was performed by comparative analysis between the sequences obtained for the sample under examination and the normal sequences of the investigated genes deposited in the international GeneBank database.

### 2.3. Statistical Analysis

Genes and genotype turnout were compared between the whole set of chosen genes of the two groups (COVID-19 and Healthy) by the Student *t*-test. Analysis followed, in the case of a significant result, by multiple comparisons. ACE2, Serpina3, IL-10, TNFα (tumor necrosis factor alpha), IFNγ (interferon beta) IL-1β, IL-1RN, IL-6, IL-6R, IL-10, and VDRs (vitamin D receptors, Taql, Bsml and Fokl) were considered (Figure 1).

The genotype frequencies in patients were tested with the Student *t*-test. Chi-square test was used to determine differences in the frequencies of all different genotypes between COVID-19 patients/healthy and between a set of genes and different genotypes. The odds ratio (OR) and 95% confidence interval (95% CI) and *p* value (*p* > 0.05) were calculated for disease susceptibility and clinical subtypes in relation to the studied gene polymorphism-SNP. The ANOVA test and analysis of variance were used to compare numeric variables within the two groups (IL6, vitamin D, and arterial blood gas data of COVID-19/Healthy groups) with the variability between groups, depending on the distribution of the data. Correlations were conducted to show the relation between quantitative variables using Pearson’s coefficient.

## 3. Results

The SNPs and plasma analyses (ABG, IL-6 and Vitamin D) of healthy (blue color) and COVID-19 groups (orange color) showed specific traits suggestive of either susceptibility or protective features. The ACE’s genotype susceptibilities were ID and DD genotypes (53% vs. 40% and 48% vs. 42%), while II (15% vs. 10%) showed a substantial protective trait (95% CI, *p* < 0.05) (Figure 1A). The following genotypes were considered: the Serpina3 G/T genotype was considered (61% affected vs. 53% healthy) (95% CI, *p* < 0.05) and the G/G was considered a protective genotype (21% healthy vs. 15% affected) (Figure 1B); th CRP G/G was considered a protective genotype (58% healthy vs. 46% affected) (95% CI, *p* < 0.05) (Figure 1C); VDR Fok1 revealed the T/C genotype as a higher risk in infection (56% vs. 44%) (95% CI, *p* < 0.05) and the T/T as a protective genotype (40% vs. 29%) (95% CI, *p* < 0.05) (Figure 1D); VDR Bsm1, the C/C genotype, showed higher prevalence in the COVID-19 patients (49% vs. 44%) (Figure 1E); VDR Tak1, the G/G genotype, was shown to be protective against the infection (37% vs. 27%) (95% CI, *p* < 0.05) (Figure 1F); IL1β rs16944, the T/T genotype, was shown to exert a protective activity (9% vs. 2%, no statistical significance), while the C/T genotype indicated a kind of susceptibility to the disease (56% vs. 49%) (95% CI, *p* < 0.05) (Figure 1G,H); IL6 rs1800796, the G/G genotype, showed a slight risk in the disease (88% vs. 83%) (95% CI, *p* < 0.05) (Figure 1I); IL6 rs1800795572, the G/G and G/C genotype, showed higher prevalence in the COVID-19 patients, respectively (71% vs. 33%) and (57% vs. 24%) (95% CI, *p* < 0.05), while the C/C polymorphism indicated a sort of protection (14% vs. 5% no statistical significance) (Figure 1L); the IL10 A/A genotype, which indicate a low gene expression functionality, confirmed a higher prevalence in the COVID-19 patients vs. healthy ones (27% vs. 9%) (95% CI, *p* < 0.05), while the genotype G/A showed a higher rate of protection against the infection (56% vs. 39%) (95% CI, *p* < 0.05) (Figure 1M); IL1RN, of the genotype C/C, confirmed the protective function of the gene against the infection (14% vs. 2% no statistical significance), while the genotypes C/T (normal genotype) and T/T (down-expression) were suggestive of higher prevalence in the COVID-19 patients, respectively (59% vs. 53%) and (39% vs. 33) (95% CI, *p* < 0.05) (Figure 1N); IL6R, of the genotype A/A, was suggestive of lower expression and higher prevalence in the COVID-19 patients (41% vs. 33%) (95% CI, *p* < 0.05) (Figure 1O); the IFNγ A/A low expression genotype indicated a higher prevalence in the COVID-19 patients (34% vs. 16%), while the A/T genotype (normal) showed a higher protection against the infection (69% vs. 39%) (95% CI, *p* < 0.05) (Figure 1P); TNFα, the GG polymorphism, indicated low expression and confirmed a higher prevalence in the COVID-19 patients (83% vs. 74%) (95% CI, *p* < 0.05) (Figure 1Q). All analyzed SNPs were reported in Table 2.

The plasma level of vitamin D and IL6 (D low and IL6 high in COVID-19 patients; D high and IL6 low in healthy patients) and ABG were highly distinctive patterns between the COVID-19 and healthy groups. The correlation analysis was conducted by using Pearson’s r (to quantify the strength of the relationship between variables) to see if variables were significantly related to healthy/COVID-19 patients. There was a statistically significant difference between COVID-19 patients compared to control healthy subjects as regards the mean values ± SD of the whole investigated parameters (*p* < 0.001 each). The most affected age group was 51–80 years among males and above 60 years among females. The most common ABG finding was high pH, indicating alkalosis, found among 41 patients (N26 = 58.3%) patients. Acidosis was rare and was seen in only one (2.43%) patient. Statistically significant correlation was found between PaO2, PaCO2, and pH in relation to vitamin D and IL6 (Pearson correlation coefficient (r) = −0.153, *p* = 0.007) (Table 3, Table 4, Table 5, Table 6, Table 7 and Table 8; Figure 2, Figure 3, Figure 4, Figure 5, Figure 6 and Figure 7). ANOVA variables were analyzed from data by comparing the variability within these groups to the variability between groups to evaluate if significant differences were present between the two groups.

## 4. Discussion

Diverse aspects have been described for COVID-19 disease considering risk factors, such as age, gender, and metabolic preexisting condition, such as diabetes, obesity, and hypertension. However, genetic background has been frequently ignored to be eventually confirmed as a major player in COVID-19 infection. Different univariate approaches on genotypes were seen to be correlated with COVID-19 susceptibility, either in mortality or immunity. In the present study, patients who were positive for SARS-CoV-2 infection showed a variable degree of symptomatology. In addition, it was quite common to see people from the same family or from the same working environment to be either mild-severely affected or completely untouched by the infection. Their outcomes were eventually confirmed by the negative results from RT-PCR test.

The IL6 and vitamin D levels, together with ABG markers, were the keys used to predict the course of the infection. Nonetheless, despite the results and the recent outcomes (2022), these issues are still substantially based on hypotheses from clinical, epidemiological, and pathophysiological observations based on current immunological standpoints [5].

We assumed that age and preexisting metabolic conditions in COVID-19-affected patients had to be related to ABG’s discrepancies and disease worsening. In a previous study on COVID-19, our team showed that the most common disturbances seen in admitted patients were observed at the level of the acid-base balance and alveolar gas-exchanges (O_2_–CO_2_), predominantly made of respiratory alkalosis, hypocapnia, and hypoxia, despite the grade of infection severity [5].

Variables were described as medians and categorical variables were described as percentages and frequencies. The Spearman correlation test was conducted for the calculation of correlation between pH and other ABG parameters. There was a highly statistically significant difference between COVID-19 patients compared to control healthy subjects of the whole investigated parameter (*p* < 0.001 each). The most affected age group was 51–80 years among males and above 60 years among females. The most common ABG finding was high pH, indicating alkalosis, found among 41 patients (58.3%). Acidosis was rare and was seen in only one patient (2.43%). Statistically significant correlation was found between PaO_2_, PaCO_2_, and pH, with vitamin D and IL6 (Pearson correlation coefficient (r) = −0.153, *p* = 0.007) being considered.

In our present study, among the mildly to severe affected survivors (>96%), age and metabolic syndrome (MS) (hypertension, hypercholesterolemia, being overweight, and diabetes type 2) was the most common comorbidity. In MS, the level of vitamin D declines while inflammatory processes tend to increase, resulting in disrupted bones, nerves, kidneys, and heart homeostasis. The crucial immune regulator function of vitamin D in both the innate and adaptive immune system was demonstrated by the discovery of the presence of VDR expression in almost all cells of the immune system, as well as the presence of the metabolizing hormones in immune cells. In addition, it is also well confirmed that the protective and immune modulatory role of the gut epithelial VDR of the mucosal barrier and gut innate immunity are important [52,53].

The outcomes, here reported, in regard to a healthy patient group which received vitamin D supplements during the September–November 2021 pandemic period (5.000–10.000 IU per day, per person), highlighted the overall vitamin D antagonizing activity against Sars-CoV-2 infection (none of the individuals involved during that specific period was infected). These preliminary outcomes are confirmed by a previous study in which it was shown that a low level of vitamin D was often found together with a reduced functionality of renal activity (low eGFR), and both were seen to be concomitant with an increased mortality risk in patients with lung affections and heart failure (HF) in COVID-19 patients [53].

To better clarify the situation, we parallelly investigated the possibility of a natural predisposition to the disease by analyzing fifteen SNPs carried out in 10 genes. The overall outcomes showed the following: ACE-1 (I/D higher prevalence in COVID-19 group), Serpina3 (G/T higher prevalence in COVID-19 group), CRP (G/G higher prevalence in healthy group), IL6 rs1800795 (G/G-G/C higher prevalence in COVID-19 group), and IL10 (G/A higher prevalence in healthy group; A/A higher prevalence in COVID-19 group) and IL1RN (C/T-T/T higher prevalence in COVID-19 group; C/C higher prevalence in healthy group), IL6R (A/A lower prevalence in COVID-19 group), VDR (Fok1 TC higher prevalence in COVID-19 group, and T/T higher prevalence in the healthy group; Taq1 A/G higher prevalence in COVID-19 group, G/G higher prevalence in healthy group), IFNγ (A/A lower prevalence in COVID-19 group, A/T higher prevalence in healthy group), and TNFα (G/G higher prevalence in COVID-19 group).

Many of these genes with mutations were seen to be correlated with an elevated risk of mortality in COVID-19. Polymorphisms in the above-mentioned genes encode for specific molecules and proteins implicated in the regulatory mechanism of immune responses and in final processing antigens for presentation through MHC-I [52].

Therefore, we proposed to include these polymorphisms into a model with age, sex, blood count, and inflammatory parameters (IL6, vitamin D and ABG) to obtain a clearer clinical picture of the great variability in disease severity, allowing a prediction of most severe cases. Our data have added to considerable information on different gene expressions and their deep interrelation in COVID-19 pathogenesis, indicating a complex scenario in which different genotypes or alleles of the same gene would exert different effects and functions against the virus.

For instance, while the IL6 174 G/G genotype would indicate an uncontrollable IL6 expression, the G/C is mainly related to a moderate IL6 expression. In addition, it is known that IL6 may become completely unmanageable in the presence of IL10 and IL6R genes down-expression, indicated by SNPs with genotype A/A, which is a typical scenario of the SARS-CoV-2 “cytokine storm”. Thus, SNPs’ analysis can strengthen the evidence for causal inference, as it would eventually explain the diverse degree of infection and the differences in individual’s responses to Sars-CoV-2 [52,53].

That said, susceptibility to a certain disease should be seen as multi-layered condition in which multiple factors and different polymorphisms all contribute to the final responses. Despite this, the increasing of robust data identifying genetic determinants involved in immunity move from bench to bedside in infectious diseases. The case of COVID-19 will certainly require a more comprehensive approach [52,53,54,55,56,57,58].

In addition, reconsideration of our perspective on SARS-CoV-2 would certainly result in reviewing its particular pleiotropic behavior in response to the human species adaptation process. Such unsolved issues in COVID-19 are presented under the protein–homeostasis system hypothesis, wherein every disease, including COVID-19, would be the result of integrated responses by the host immune system towards pathogen or exogeneous biochemical properties [53,54,55,56,57,58,59,60].

We are well aware of the limitations of this study, as the obtained results would require further validation in larger cohorts with different ethnicities and geographical regions. In addition, due to the limited number of participants, we could not control a few basic confounding factors. For example, the SNP presence of patient cohorts was difficult to weigh, especially considering the immense difficulties of the period. Second, although our data showed, somehow, an underlying association between a genetic predisposition to COVID-19 susceptibility and severity, it should not be mistaken that those small cohorts could be reductive once applied to the general population. Third, the COVID-19 disease is a binary exposure that could introduce unexpected bias, which may mitigate the causal association between the genetic predisposition and the increased COVID-19 severity. For patients unaffected by the COVID-19 disease through active exposure, an intensified management and surveillance for one’s immunity profile is still significant.

## 5. Conclusions

COVID-19 has been generally accepted as a multigenic and multifactorial disease with different determinants. The identification of the factors implicated in the infection by SARS-CoV-2 is the key to better understand the etiology and physiopathological mechanisms of COVID-19 individuals’ infection prevention, progression, and treatment management. The results of the present study, despite their intrinsic limitations, showed that many different gene SNPs could be associated with a higher risk for COVID-19 infection. To conclude, the overall understanding of an individual’s specific polymorphisms might help to better explain COVID-19 outcomes when trying to promote the genetic profiling for setting up personalized therapies to improve COVID-19 treatment strategies.

## Figures and Tables

**Figure 1 diagnostics-12-02824-f001:**
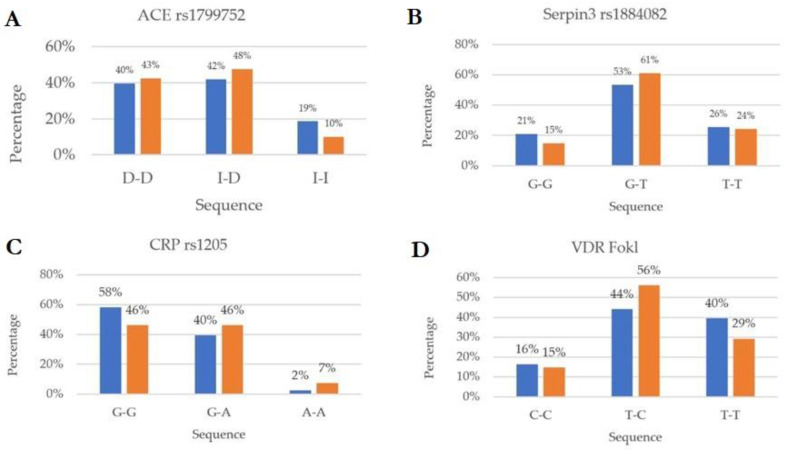
Genotype distribution of 15 SNPs among the healthy (blue) and affected (Orange) subjects analyzed. (**A**) SNPs and ACE: the analysis indicates that the I/I genotype is suggestive of higher protection to COVID-19 infection (19% vs 10%), while the expression of I/D may represent a higher susceptibility to the infection; a higher frequency of the I/D genotype (48% vs. 42%) and the D allele (>43% overall) was reported in COVID-19 patients than controls (95% CI, *p* < 0.05) [22]. (**B**) SNP in the Serpina3 (AACT) gene seems to be tissue-specific and influences protease targeting considered to play a key role in neuro-degenerative disorders. The gene is involved in expressing the protein during acute and chronic inflammation. The Serpina3 seems to be significantly associated with critical illness in COVID-19. The analysis indicates that the G/G genotype is suggestive of higher protection to COVID-19 infection (21%vs 15%; 95% CI, *p* < 0.05) [23,24,25,26]. (**C**) Circulating CRP concentration reflects systemic inflammation. Some polymorphisms have been linked with high expression during inflammatory conditions such as in COVID-19; the outcomes indicated that the G/G genotype is suggestive of higher protection to COVID-19 infection (46% vs. 40%), while the expression of G/A (normal genotype) (95% CI, 95% CI, *p* < 0.05) and A/A (7% vs. 2%) (gene down-expression) may represent a sort of susceptibility to the infection; in allelic comparison, the G allele was strongly associated with COVID-19 infection (95% CI, 95% CI, *p* < 0.05) [27,28,29]. (**D**–**F**) VDR gene polymorphisms might play critical roles in the vulnerability to infection and severity of COVID-19. The allele frequencies for the SNPs of the VDR gene (*FokI*, *BsmI*, and *TaqI*) were compared between infected cases and controls. The analysis regarding VDR *Fok1* indicates that the T/C genotype might be significantly associated with COVID-19 infection (56% vs. 44%; 95% CI, *p* < 0.05), while the expression of T/T (40% vs. 29%) indicated a significant protection against the infection (95% CI, *p* < 0.05); *Taq1*, the A/G genotype, was significantly associated with COVID-19 infection (59% vs. 47%; 95% CI, *p* < 0.05), while the expression of G/G (37% vs. 27%) indicated a significant protection against the infection (95% CI, *p* < 0.05) [49,50]. (**G**,**H**) the carriers with IL1β c-81 with C/T genotype has been associated with increased IL-1β production and grade of inflammation. The carriers with 511 C/T genotypes were also seen with higher incidence of COVID-19 infection (95% CI, *p* > 0.05) [39,40,41]. (**I**,**J**) The human IL-6 572 gene is involved in anti-E production, with the allele G as the major risk allele. This is evidence that there is a clear genetic influence on plasma levels of fibrinogen and CRP. However, the data did not indicate a clear involvement in COVID-19 disease. Conversely, IL-6 174 rs1800795 polymorphism has been shown to be highly related to inflammatory and neuropathic patterns in several diseases such as DM2, atherosclerosis, CVD, and COVID-19. The G/G genotype is suggestive of IL-6 gene overexpression. There were significant differences between the two groups (COVID-19 71% vs. 33% healthy) (95% CI, *p* < 0.005); conversely, the G/C seemed to exert a more protective activity (53% vs. 24%; 95% CI, *p* < 0.05) [33,34,35]. (**K**) The *IL10* (rs1800896) gene polymorphism that compromises the expression of IL10 is implicated in susceptibility to pulmonary infection and general inflammatory states, typical of tuberculosis (PTBC) and ARDS, specifically in adult and elderly. Data showed that, among patients with ARDS, the -*1082GG* genotype with normal expression of IL10 revealed a decrease in severity with lower systemic failures and lower mortality rates. There was a mild significant correlation between the frequency of the A/A genotype and the prevalence of COVID-19 cases recorded in this study (27% vs. 9%) (95% CI, *p* < 0.05). Conversely, there was a significant correlation between the frequency of the G/A genotype and the prevalence of a protection against COVID-19 (56% vs. 39%) (95% CI, *p* < 0.05) [30,31,32]. (**L**) The IL1 receptor antagonist (*IL1RN*) belongs to the IL1 family, some of the gene variants were showed to be involved in several inflammatory degenerative conditions and high susceptibility to infection. In fact, the SNP expression of IL1RN C/T (59% vs. 53%) and T/T (39% vs. 33%), both gene down-expression genotypes, showed a higher susceptibility to COVID-19 infection, with allele T playing the major role (95% CI, *p* < 0.05). On the other hand, the SNP with the C/C genotype (14% vs. 2%) was seen to increase the serum level of IL-1 RN, which blocks the action of IL-1 (no statistical significance) [42]. (**M**) The IL-6 receptor antagonist (*IL6R*) belongs to the IL-6 family. The gene variants were shown to be quite ubiquitous, and some variants were seen to be involved in several inflammatory degenerative conditions and high susceptibility to infection. Some others were conversely revealed to exert a protective function. In fact, the SNP expression of the IL-6R gene with A/C (53% vs. 49%) and C/C (14% vs. 10%) genotypes were seen to protect against COVID-19 infection, with allele C playing the major role. Conversely, the A/A genotype (41% vs. 33%) was seen significantly toward the infection (95% CI, *p* < 0.05). Similarly, to IL1RN these A/C and C/C genotype seems to exert a protective function in decreasing the risk of development of acute respiratory distress syndrome and improves survival from septic shock, which are the two main causes of ICU admission and mortality in COVID-19 [12,13,14,15,16,17,36,37,38,43,44,45]. (**N**) IFNγ + 874A/T (rs2430561) gene polymorphism can increase or decrease the risk of a pathogenic infection and susceptibility for pulmonary infection such as tuberculosis, SARS, and oral infection. Data showed that, among patients with COVID-19, the IFN-γ + 874A/T genotype was higher (60% vs. 39%). In this study, there was a significant correlation between the frequency of A/A genotype (gene lower expression of IFN-γ) (34% vs. 16%) and the prevalence of COVID-19 cases (95% CI, *p* < 0.05) [47,48]. (**O**) The TNF-α -308 G > A gene is associated with high risk level of CVD, atherosclerosis, and high susceptibility to pulmonary disease risk. After genotype testing, a statistically significant difference between the patients and controls was found in regards to the genotype distribution, where the G allele was more expressed in patients vs. controls with (G/G 83% vs. 74%; 95% CI, *p* < 0.05). This indicated the G allele (G/G and G/A) as more susceptible to the disease (95% CI, *p* < 0.05) [44,45,46].

**Figure 2 diagnostics-12-02824-f002:**
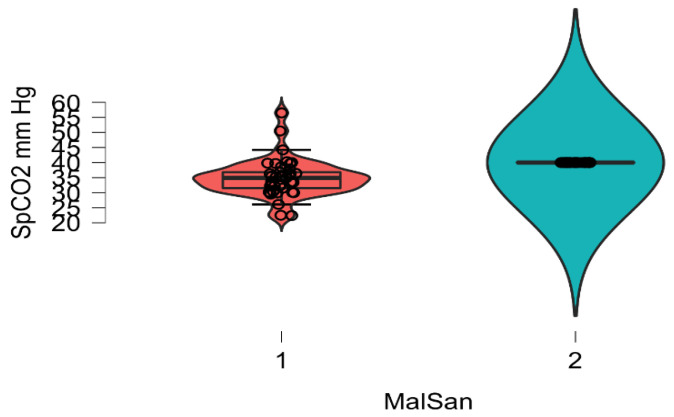
Boxplots of SpCO_2_ in mm/Hg concentrations in COVID-19 patients (red) and healthy control (blue) cases stratified by SpCO_2_ concentration. Of note, COVID-19 status (*p* = 0.005) and SpCO_2_ (*p* < 0.001) are significantly associated with clinical values in a multivariable linear regression analysis of sex, age, and COVID-19 status.

**Figure 3 diagnostics-12-02824-f003:**
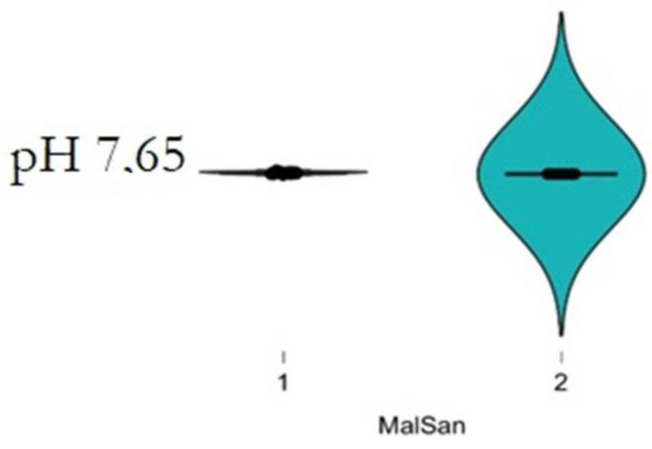
Boxplot representation of pH distribution in the first day after the admission patients (1) compared with healthy (2) stratified by pH concentration. Of note, COVID-19 status (*p* = 0.005) and pH (*p* < 0.001) are significantly associated with clinical values in a multivariable linear regression analysis of sex, age, and COVID-19 status.

**Figure 4 diagnostics-12-02824-f004:**
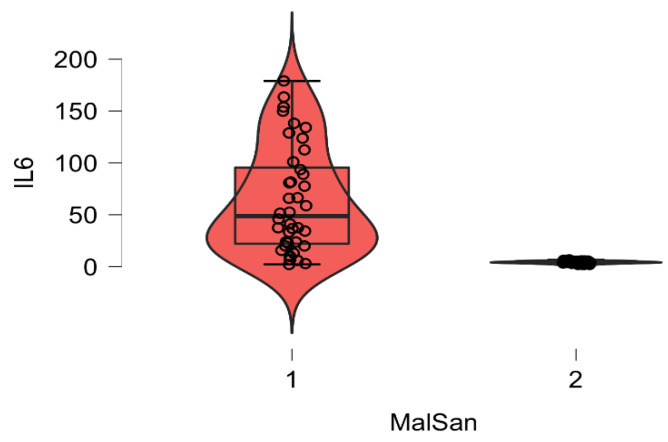
Boxplot showing the plasma concentration of IL-6 within 48 h of hospitalization in patients with COVID-19 (red) and healthy individuals.

**Figure 5 diagnostics-12-02824-f005:**
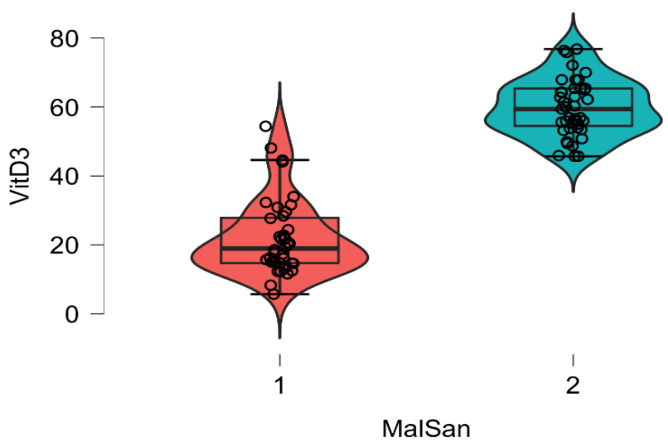
Box plot showing the plasma concentration of vitamin D within 48 h of hospitalization in patients with COVID-19 (red) and healthy individuals.

**Figure 6 diagnostics-12-02824-f006:**
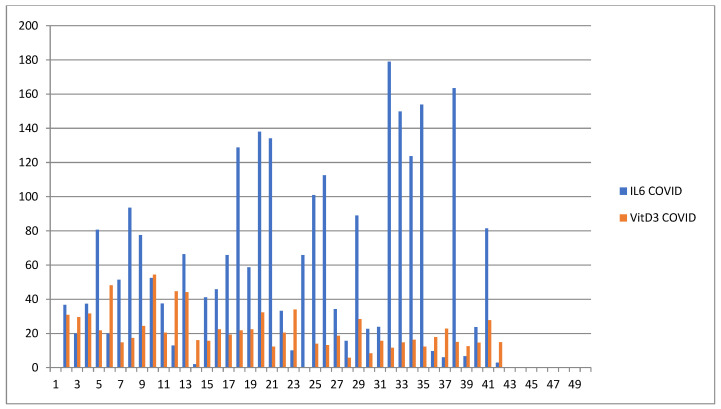
Graphic showing the plasma concentration of vitamin D (blue) and IL-6 (orange-red) within 48 h of hospitalization in patients with COVID-19.

**Figure 7 diagnostics-12-02824-f007:**
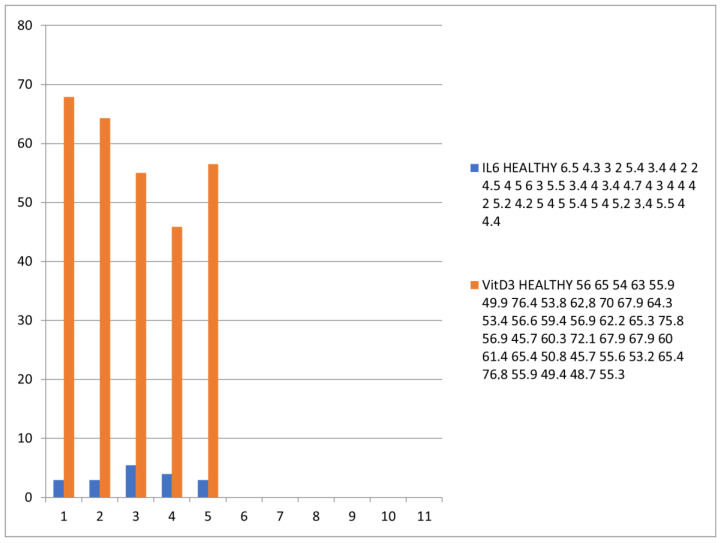
Graphic showing the plasma concentration of vitamin D (blue) and IL-6 (orange-red) in healthy individuals.

**Table 1 diagnostics-12-02824-t001:** List of analyzed SNPs.

Gene	Locus	Protein Coding	ID SNP	Variant Information		Variation Effect
*ACE 2*	Chromosome Xp22.2	Angiotensin converting Enzyme2	rs1799752	NG_011648.1:g.16471_16472ins	I/D	Intron variant
*SERPINA3*	Chromosome 14q32.1	Serpin family A member 3	rs1884082	NG_012879.1:g.4964G>T	SNV	Upstream variant
*CRP*	Chromosome 1q23.2	C-reactive protein	rs1205	NG_013007.1:g.7147G>A	SNV	3 prime UTR variant
*IL1β*	Chromosome 2q14.1	Interleukin 1 beta	rs16944	NG_008851.1:g.4490T>C	SNV	2KB upstream variant
			rs1143634	NG_008851.1:g.8967C>T	SNV	Synonymous variant
*IL1RN*	Chromosome 2q14-q21	Interleukin 1 receptor antagonist	rs419598	NG_021240.1:g.16738T>C	SNV	Synonymous variant
*IL6*	Chromosome 7p15.3	Interleukin 6	rs1800796	NG_011640.1:g.4481G>C	SNV	Non-coding Transcript Variant
			rs1800795	NG_011640.1:g.4880C>G	SNV	Intron variant
*IL6R*	Chromosome 1q21.3	Interleukin 6 receptor	rs2228145	NG_012087.1:g.54302A>C; NG_012087.1:g.54302A>T	SNV	Missense variant
*IL10*	Chromosome 1q32.1	Interleukin-10	rs1800896	NG_012088.1:g.3943A>G	SNV	Intron variant
*TNFα*	Chromosome 6p21.3	Tumor necrosis factor alpha	rs1800629	NG_007462.1:g.4682G>A	SNV	2KB upstream variant
*IFNγ*	Chromosome 12q15	Interferon gamma	rs2430561	NG_015840.1:g.6000A>T	SNV	Intron variant
*VDRs*	Chromosome 12q13.11	Vitamin D receptor	rs731236 *	NC_000012.12:g.47844974A>G	SNV	Initiator codon variant
		Vitamin D receptor	rs2228570 **	NG_008731.1:g.30920T>C	SNV	Initiator codon variant
		Vitamin D receptor	rs1544410 ***	NC_000012.12:g.47846052C>T	SNV	Intron variant

I/D: insertion/deletion; SNV: Single Nucleotide Variation. * Taq1 polymorphism; ** Fok1 polymorphism; *** Bsm1 polymorphism.

**Table 2 diagnostics-12-02824-t002:** SNPs and genotypes distribution among affected (A; n = 41) and healthy (H; n = 43) subjects; all SNPs are from in inflammatory and regulatory responses.

VDR																																												
ACE2			FokI			Bsm1			TagI			Serpina3			CRP			IL1β						IL6						IL10			IL1RN			IL6R			IFNγ			TNFα		
rs1799752												rs1884082			rs1205			rs16944			rs1143634			rs1800796			rs1800795			rs1800896			rs419598			rs2228145			rs2430561			rs1800629		
	A	H		A	H		A	H		A	H		A	H		A	H		A	H		A	H		A	H		A	H		A	H		A	H		A	H		A	H		A	H
D/D	17	17	T/T	12	17	T/T	4	5	A/A	6	7	G/G	6	9	G/G	19	25	C/C	17	18	C/C	21	23	G/G	34	39	C/C	1	6	A/A	14	15	T/T	24	23	A/A	16	14	A/A	10	7	G/G	34	32
I/D	20	18	T/C	23	19	T/C	17	19	A/G	24	20	G/T	25	23	G/A	19	17	C/T	23	21	C/T	19	18	G/C	5	3	C/G	11	23	A/G	16	24	T/C	16	14	A/C	22	23	A/T	17	26	G/A	7	7
I/I	4	8	C/C	6	7	C/C	20	19	G/G	11	16	T/T	10	11	A/A	3	1	T/T	1	4	T/T	1	2	C/C	2	1	G/G	29	14	G/G	11	4	C/C	1	6	C/C	3	6	T/T	14	10	A/A	0	4

**Table 3 diagnostics-12-02824-t003:** Descriptive analysis of the variables under examination, divided by COVID-19 and healthy groups.

Variable Descriptive Analysis							
		Valid	Missing	Average	St. Dev.	Min	Max
SpO_2_ mm Hg	COVID-19	41	0	66.62	18.71	36	124.8
	Healthy	43	0	87.50	0.00	87.5	87.5
SpCO_2_ mm Hg	COVID-19	41	0	34.87	6.25	22.4	56.5
	Healthy	43	0	40.00	0.00	40	40
pH	COVID-19	41	0	7.47	0.06	7.33	7.63
	Healthy	43	0	7.40	0.00	7.4	7.4
IL6	COVID-19	40	1	63.57	51.08	2	179
	Healthy	43	0	4.08	1.12	2	6.5
VitD3	COVID-19	40	1	3.63	0.69	5.7	54.4
	Healthy	43	0	3.93	0.56	45.7	76.8

**Table 4 diagnostics-12-02824-t004:** ANOVA. Descriptive analysis related to ABG, IL6, and Vit-D3.

Variable Descriptive Analysis		
	F	Sig.
SpO_2_ mm Hg	53.55	*** 0.000
SpCO_2_ mm Hg	28.91	*** 0.000
pH	57.87	*** 0.000
IL6	55.59	*** 0.000
VitD3	291.426	*** 0.000

Note. ‘***’ <0.001. F = F test (variation between sample means/variation within the considered samples).

**Table 5 diagnostics-12-02824-t005:** Analysis of the correlation between variables and healthy vs. sick status.

Variables	Pearson’s r	*p*-Value
Healthy vs. COVID-19		
SpO_2_ mm Hg	0.631	*** 0.000
SpCO_2_ mm Hg PH	0.511 −0.643	*** 0.000 *** 0.000
IL6	−0.643	*** 0.000
VitD3	0.892	*** 0.000

Note. ‘***’ <0.001.

**Table 6 diagnostics-12-02824-t006:** Analysis of the correlation between variables and healthy vs. COVID-19 infected status (see the relevance/contribution of each factor when all together—ABG’s values and IL6 with Vit.3).

Correlation between Variables						
Variables		SpO_2_ mm Hg	SpCO_2_ mm Hg	pH	IL6	VitD3
SpO_2_ mm Hg	Pearson’s r	—				
	*p*-value	—				
SpCO_2_ mm Hg	Pearson’s r	0.252	—			
	*p*-value	** 0.021	—			
pH	Pearson’s r	−0.448	−0.548	—		
	*p*-value	*** 0.000	*** 0.000	—		
IL6	Pearson’s r	−0.378	−0.212	0.626	—	
	*p*-value	*** 0.000	0.056	*** 0.000	—	
Vit. D3	Pearson’s r	0.063	−0.053	−0.087	−0.163	—
	*p*-value	0.572	0.636	0.435	0.143	—

Note. ‘**’ <0.01, ‘***’ <0.001.

**Table 7 diagnostics-12-02824-t007:** Analysis of linear regressions.

Model		Sum of Squares	df	Mean Square	F	*p*
H_1_	Regression	17.200	5	3.440	75.190	*** 0.000
	Residual	3.523	77	0.046		
	Total	20.723	82			

Note. ‘***’ <0.001.

**Table 8 diagnostics-12-02824-t008:** Analysis of the coefficients.

Coefficients						
Model		Unstandardized	Standard Error	Standardized	t	*p*
H_0_	(Intercept)	1.518	0.055		27.511	*** 0.000
H_1_	(Intercept)	1.009	6.386		0.158	0.875
	SpO_2_ mm Hg	0.004	0.002	0.140	2.001	** 0.049
	SpCO_2_ mm Hg	0.010	0.007	0.103	1.564	0.122
	pH	−0.123	0.837	−0.014	−0.147	0.884
	IL6	−3.788 × 10^−4^	8.923 × 10^−4^	−0.035	−0.425	0.672
	VitD3	0.017	0.002	0.728	11.051	*** 0.000

Note: ‘**’ <0.01, ‘***’ <0.001. H₀ = the null hypothesis; H₁ = the alternative hypothesis.

## Data Availability

All experimental data to support the findings of this study are available contacting the corresponding author upon request.

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
