# Peer review of "Analysis of Gene Single Nucleotide Polymorphisms in COVID-19 Disease Highlighting the Susceptibility and the Severity towards the Infection"

_diagnostics, 2022, doi:10.3390/diagnostics12112824_

Round 1

Reviewer 1 Report

Thank you for possibility to review an interesting study by Tomassone et al. The paper points out an important issue of genetic predisposition to COVID-19 infection. 

I have some comments and questions

1. It is much unclear what kind of population formed the “healthy” group. Were they healthy volunteers or patients with symptoms of infection in whom COVID-19 infection was excluded after PCR test and CT? 

In my humble opinion, you should not treat as healthy control people who AT THE MOMENT do not suffer from COIVD-19 but could have had in the past (it is not clarified in the text). Only those who never suffered from COVID-19 despite of contact with the virus may be treated as control group. 

Please clarify this issue. 

2. line 137 contains a part of sentence without the end “Polymorphisms of the”

3. line 255 “perfermode” – plesase correct

4. line 257 – Quality and, as I suppose, also quantity were measured 

5. Statistical analysis – the frequencies cannot be evaluated with t-test

6. Did the authors evaluate Hardy-Weinberg equilibrium?

7.Results section – lines 298-317 - contains percentages without p-values which make the reading difficult and not clear. I found the p-values in the description of Figure 1. This is however very long and maybe the authors would consider moving some of this large text to the main text as it difficult to read. Moreover, p values are described as significant while presented as p>0.05. it would be clearer to show p-values in particular figures. 

Figure 1A – percentage for healthy is 101%, please correct. Add explanation of colors at the beginning or for each figure. 

8. Table 2 is cut and therefore not clear. 

9. For each SNP, the GenBank shows the frequencies in population. Did the authors consider using/comparing or at least discussing it?

10. Line 333 – expression is used in the other context

11. line 407 correct 1 patients into 1 patient

12. please clarify explanation for Table 4

13. Clarify abbreviations in tables

14. lines 466 – 468 – These data were not presented in the Results, thus authors should not discuss them, similarly, lines 476-477 – what led the authors to this statement. 480 – infection severity was not analyzed in the study 

15. Line 488 I cannot find the significance of correlation for vit D in theResults.

16. Some of the text from the Discussion should be positioned in the Results. You should discuss yours and other authors’ results and not present them in the Discussion. 

17. Conclusion section contains description of limitations which should be positioned at the end of the Discussion. Please re-write it and write the conclusion. 

Nevertheless, my major concern is comparison of COVID-19 infected patients with a group of people who AT THE MOMENT do not suffer from COVID-19 disease. We cannot exclude their potential for infection in another period. Therefore, if you not convince me better, it seems that the study design is incorrect. 

Author Response

Comments and Suggestions for Authors

Thank you for possibility to review an interesting study by Tomassone et al. The paper points out an important issue of genetic predisposition to COVID-19 infection. 

I have some comments and questions

  1. It is much unclear what kind of population formed the “healthy” group. Were they healthy volunteers or patients with symptoms of infection in whom COVID-19 infection was excluded after PCR test and CT? 

Thanks the Reviewer for the comment. The control group was composed by healthy individuals which were negative to RT-PCR COVID-19 test without clinical symptoms of lung infection

In my humble opinion, you should not treat as healthy control people who AT THE MOMENT do not suffer from COIVD-19 but could have had in the past (it is not clarified in the text). Only those who never suffered from COVID-19 despite of contact with the virus may be treated as control group. 

The reviewer is absolutely right. However, the study, as well underlined in the test, was conducted in the period of the second wave of the COVID-19 pandemic between September - November 2020, a period marked not only by a severe lock-down but also with a tremendous amount of work especially for us that we were in Emergency Unit. The main intent was to understand the intrinsic dynamic of the disease, therefore we focused on potential effect of specific gene polymorphisms. This article it’s just a part of a wider research effort conducted in a small suburban hospital in Southern Italy. With hindsight it is clear that more and better could have been done but again our priority at that time was to make better assessment of the virus. The patients in the control group were patients who received an anti-inflammatory therapy consisting of Vitamin D, Vitamin K2 and aspirin. From our data that have not been published, during that period none of those patients were hospitalized for COVID-19 infection even  someone in a later period became mildly infected (2. Materials and methods 2.1. Study Design).

Please clarify this issue. 

  1. line 137 contains a part of sentence without the end “Polymorphisms of the”

We have corrected.

  1. line 255 “perfermode” – plesase correct

We have corrected.

  1. line 257 – Quality and, as I suppose, also quantity were measured 

We have corrected.

  1. Statistical analysis – the frequencies cannot be evaluated with t-test

The reviewer is correct, however the term “frequencies” were referred to genes and genotype as “presence” “turn-out”.

  1. Did the authors evaluate Hardy-Weinberg equilibrium?

No we did not because the using of T-Test and Anova was considered sufficient for the our case.

7.Results section – lines 298-317 - contains percentages without p-values which make the reading difficult and not clear. I found the p-values in the description of Figure 1. This is however very long and maybe the authors would consider moving some of this large text to the main text as it difficult to read. Moreover, p values are described as significant while presented as p>0.05. it would be clearer to show p-values in particular figures. 

Figure 1A – percentage for healthy is 101%, please correct. Add explanation of colors at the beginning or for each figure. 

Thanks to the Reviewer for the valuable comment, the P value was corrected all over the main text and figure 1 was organized better.

  1. Table 2 is cut and therefore not clear. 

The table 2 has been corrected however it will depend on editor to use the right format.

  1. For each SNP, the GenBank shows the frequencies in population. Did the authors consider using/comparing or at least discussing it?

Yes we did (Genoma lab is completely based on GenBank data), however not all gene analyzed for this study were at that time linked to COVID-19. This study was for that period quite innovative in term of complexity involved in the disease.

  1. Line 333 – expression is used in the other context

Correction has been made.

  1. line 407 correct 1 patients into 1 patient

Correction has been made.

  1. please clarify explanation for Table 4

The clarification has been added

  1. Clarify abbreviations in tables

The clarification has been added

  1. lines 466 – 468 – These data were not presented in the Results, thus authors should not discuss them, similarly, lines 476-477 – what led the authors to this statement. 480 – infection severity was not analyzed in the study.

Thanks to the Reviewer for the valuable comment. The data that we have indicated in the paragraph are part of previously published work (ref.5)

Balzanelli, M.; Distratis, P.; Catucci, O.; Amatulli, F.; Cefalo, A.; Lazzaro, R.; Aityan, K.S.; Dalagni, G.; Nico, A.; De Michele, A.; Mazza, E.; Tampoia, M.; D'Errico, P.; Pricolo, G.; Prudenzano, A.; D'Ettorre, E.; Di Stasi, C.; Morrone, L.; Nguyen, K.; Pham, H.V.; Inchingolo, F.; Tomassone, D.; Gargiulo Isacco, C. Clinical and diagnostic findings in COVID-19 patients: An original research from SG Moscati Hospital in Taranto Italy. Journal of biological regulators and homeostatic agents 2021, 35, 171–183.

  1. Line 488 I cannot find the significance of correlation for vit D in the Results.

The D, IL6 and ABG outcomes were mentioned in the results section. The plasma level of Vitamin D and IL6 (D low and IL6 high in COVID-19 patients; D high and IL6 low in healthy patients) and ABG were highly distinctive patterns between the COVID-19 and Healthy groups. The correlation analysis was conducted by using Pearson’s r (quantify the strength of the relationship between variables) to see if variables were significantly related to Healthy/COVID-19. There was a statistically significant difference between COVID-19 patients compared to control Healthy subjects as regards the mean values ± SD of the whole investigated parameters (P < 0.001 each).

  1. Some of the text from the Discussion should be positioned in the Results. You should discuss yours and other authors’ results and not present them in the Discussion. 

We really appreciated the reviewer's suggestions. The citation of different authors in the Discussion section was essential to explain better our point of view. The citation refers to a publication of ours (ref.5).

  1. Conclusion section contains description of limitations which should be positioned at the end of the Discussion. Please re-write it and write the conclusion. 

We really appreciated the reviewer's suggestions. The adjustments were made following the theme.

Nevertheless, my major concern is comparison of COVID-19 infected patients with a group of people who AT THE MOMENT do not suffer from COVID-19 disease. We cannot exclude their potential for infection in another period. Therefore, if you not convince me better, it seems that the study design is incorrect. 

The Reviewer's decision is legitimate. However, we should consider the time frame and period this study was performed (September - November 2020). Around the world, including us of course, we have all been taken by surprise and thus completely disarmed against the lethal behavior of SARS-CoV-2. At that time we were in the phase of understanding what we were facing to. Therefore, allow us at least the ”Intuition” among many scientists and researchers, of having considered the involvement of genes and polymorphisms in the progress of the disease, we were probably the one of the few of talking about “Happy Hypoxia”. If we read this article with this perspective (perspective of Emergency Medicine) the article would possibly take its “place” in the scientific debate.  

Reviewer 2 Report

Abstract

-          The authors are encouraged to use a consistent form of the word makeup or make-up.

-          Double-check and revise COVID-1919 in line 36.

-          In the concluding sentence, “Though more research is needed, these findings may give a better representation of virus pleiotropic activity and its relation to own immune system.”, the authors did not offer any concrete results that properly represent their findings. Try to use more specific and quantifiable, if possible, sentences.

Keywords

-          Providing 12 keywords is considered overly excessive. Please revise the use of keywords and try to keep it up to 10, at most.

-          It is also recommended to list them according to their alphabetical order.

Introduction

-          In line 70, revise “Semintensive”.

-          Revise “Though the almost the totality were successfully treated all revealed persisting health problems even after the infection (long-COVID-19).”

-          Revise “The considered genes were the follow…”

-          In line 80, add the missing parenthesis “IL-1β rs16944; rs1143634)”

-          Revise “…SARS-CoV-2 infection the present intent was eventually …”

-          Revise “…The Serpina3 gene play an…”

-          In line 128, revise “…in COVID-19Plasma levels..”

-          In lines 133 and 134, since you’re specifically discussing only one gene polymorphism, then you should revise the sentence to address a singular entity.

-          In line 150, polymorphism or polymorphisms? If polymorphism, you should address the comment in the previous point.

-          In lines 169 and 170, you should clear out the ambiguity by revising the sentence structure.

-          In line 208, revise “…and suppress the transcription…”

Materials and methods

-          In line 236, “… males n = 39..” is it 39 or 29?

-          Were there any applied criteria for gender selection or, at least, getting a closer match for both groups’ genders? If not, it limits evaluation accuracy. The authors are encouraged to address this issue or at least comment on it.

Results

-          The table 2 caption is missing.

-          The authors reported their numerical results in nearly one sentence. Please consider restructuring the first paragraph of the Results.

-          The caption of Figure 1 is overly lengthy. The authors are supposed to merge the descriptive details into the Results paragraphs.

Discussion

-          In line 520, revise “…will certainly requires…”

-          Restructure the sentence in lines 522 and 523.

Conclusion

-          Focused nearly on the limitations.

-          It does not properly restate the key findings.

-          It should be further expanded.

-          The limitations should also be justified.

Core questions

-          Do you think it’s wise to conduct an extremely limited retroactive study to investigate the stated factors?

-          Assuming the justification of the study, what about COVID-19 mutants during the elapsed period?

-          Have you considered benchmarking your study to prove its merit? This is a vital key for the appraisal; without it, it won’t easy to approve your outcomes. 

Author Response

Comments and Suggestions for Authors

Abstract

-          The authors are encouraged to use a consistent form of the word makeup or make-up.

Thanks the Reviewer’s help correction has been made.

-          Double-check and revise COVID-1919 in line 36.

Thanks the Reviewer’s help correction has been made.

-          In the concluding sentence, “Though more research is needed, these findings may give a better representation of virus pleiotropic activity and its relation to own immune system.”, the authors did not offer any concrete results that properly represent their findings. Try to use more specific and quantifiable, if possible, sentences.

We really appreciated the reviewer's suggestions. The adjustments were made following the theme. However, let me add something to better explain our position.The Reviewer's decision is legitimate. We should consider the time frame and period this study was performed (September - November 2020). Around the world, including us of course, we have all been taken by surprise and thus completely disarmed about the lethal behavior of SARS-CoV-2. At that time we were still in the understanding phase what we were facing to. Therefore, allow us at least the predictive ”Intuition” among many scientists and researchers, of having considered the involvement of genes and polymorphisms in the progress of the disease, we were probably the one of the few of talking about “Happy Hypoxia”. If we read this article with this perspective (perspective of Emergency Medicine) the article would possibly take its “place” in the scientific debate.

Keywords

-          Providing 12 keywords is considered overly excessive. Please revise the use of keywords and try to keep it up to 10, at most.

-          It is also recommended to list them according to their alphabetical order.

Thanks the Reviewer’s help correction has been made.

Introduction

-          In line 70, revise “Semintensive”.

Semi-Intensive Care Unit has been corrected

-          Revise “Though the almost the totality were successfully treated all revealed persisting health problems even after the infection (long-COVID-19).”

Sentence has been corrected

-          Revise “The considered genes were the follow…”

Sentence has been corrected

-          In line 80, add the missing parenthesis “IL-1β rs16944; rs1143634)”

Sentence has been corrected

-          Revise “…SARS-CoV-2 infection the present intent was eventually …”

Sentence has been corrected

-          Revise “…The Serpina3 gene play an…”

Sentence has been corrected

-          In line 128, revise “…in COVID-19Plasma levels..”

Sentence has been corrected

-          In lines 133 and 134, since you’re specifically discussing only one gene polymorphism, then you should revise the sentence to address a singular entity.

Sentence has been corrected

-          In line 150, polymorphism or polymorphisms? If polymorphism, you should address the comment in the previous point.

Sentence has been corrected

-          In lines 169 and 170, you should clear out the ambiguity by revising the sentence structure.

Sentence has been corrected

-          In line 208, revise “…and suppress the transcription…”

Sentence has been corrected

Materials and methods

-          In line 236, “… males n = 39..” is it 39 or 29?

The number has been corrected

-          Were there any applied criteria for gender selection or, at least, getting a closer match for both groups’ genders? If not, it limits evaluation accuracy. The authors are encouraged to address this issue or at least comment on it.

Dear Reviewer the there was not criterion selection regarding the gender, all included patients were randomly involved. At that time there was no time to execute with such precision.

Results

-          The table 2 caption is missing.

Caption has been added

-          The authors reported their numerical results in nearly one sentence. Please consider restructuring the first paragraph of the Results.

-          The caption of Figure 1 is overly lengthy. The authors are supposed to merge the descriptive details into the Results paragraphs.

Thanks for the suggestion we have proceeded to adjust it

Discussion

-          In line 520, revise “…will certainly requires…”

Thanks for the suggestion we have proceeded to adjust it

-          Restructure the sentence in lines 522 and 523.

Thanks for the suggestion we have proceeded to adjust it

Conclusion

-          Focused nearly on the limitations.

We have tried to solve the issue

-          It does not properly restate the key findings.

We have restated few important points in the last part of the Discussion

-          It should be further expanded.

-          The limitations should also be justified.

Limitation were justified

Core questions

-          Do you think it’s wise to conduct an extremely limited retroactive study to investigate the stated factors?

The Reviewer's doubts are legitimate. However, we should highlight this study in the context and period in which it was performed. The intent was specifically to identify the genetic predisposing components underlying the differences identified from patient to patient. Aware of the fact that these assumptions may not be considered exclusive to the COVID-19 infection and that they belonged to other viral infections.

-          Assuming the justification of the study, what about COVID-19 mutants during the elapsed period?

We agree with the Reviewer’s assertion. As a matter of fact, later during the pandemic period we have published papers about new rapid diagnostic procedures, by RT-PCR, in which we considered in details the presence of variants and mutation:

Pham VH, Gargiulo Isacco C et al. Rapid and sensitive diagnostic procedure for multiple detection of pandemic Coronaviridae family members SARS-CoV-2, SARS-CoV, MERS-CoV and HCoV: a translational research and cooperation between the Phan Chau Trinh University in Vietnam and University of Bari "Aldo Moro" in Italy. Eur Rev Med Pharmacol Sci. 2020 Jun;24(12):7173-7191. doi: 10.26355/eurrev_202006_21713. PMID: 32633414 (patented);

Pham VH, Gargiulo Isacco C et al. Development of the Multiplex RT-Realtime PCR for detection and identification of the different variants of the current circulating SARS-COV-2 (editing).

-          Have you considered benchmarking your study to prove its merit? This is a vital key for the appraisal; without it, it won’t easy to approve your outcomes.

Yes we have considered but again at that time was difficult for us to perform that we were working with very little data, little knowledge, limited know-how and in a really tough environment with huge unknown variables ..... The rest is history that everyone knows now. 

Round 2

Reviewer 1 Report

Thank you for the adherence to my comments 

Author Response

Thanks!

Reviewer 2 Report

The followings are the minor language mistakes I quickly could detect. However, I would recommend that you give it another thorough read to detect any minor language/grammatical mistakes.

In line 503, remove the extra “how”

In line 537, revise “Said that” I believe you meant “That said”.

In line 548, revise “aware about”.. You should use “aware of”

Author Response

All corrections have been made.

Thanks!
